# FBXO24 deletion causes abnormal accumulation of membraneless electron-dense granules in sperm flagella and male infertility

Yuki Kaneda[1,2], Haruhiko Miyata[1]*, Zoulan Xu[1,2], Keisuke Shimada[1], Maki Kamoshita[1], Tatsuya Nakagawa[1,2], Chihiro Emori[1], Masahito Ikawa[1,2,3,4,5]*

[1]Research Institute for Microbial Diseases, Osaka University, Osaka, Japan; [2]Graduate School of Pharmaceutical Sciences, Osaka University, Osaka, Japan; [3]Center for Infectious Disease Education and Research (CiDER), Osaka University, Osaka, Japan; [4]The Institute of Medical Science, The University of Tokyo, Tokyo, Japan; [5]Center for Advanced Modalities and DDS (CAMaD), Osaka University, Osaka, Japan

*For correspondence:
hmiya003@biken.osaka-u.ac.jp
(HM);
ikawa@biken.osaka-u.ac.jp (MI)

Competing interest: The authors declare that no competing interests exist.

**Abstract** Ribonucleoprotein (RNP) granules are membraneless electron-dense structures rich in RNAs and proteins, and involved in various cellular processes. Two RNP granules in male germ cells, intermitochondrial cement and the chromatoid body (CB), are associated with PIWI-interacting RNAs (piRNAs) and are required for transposon silencing and spermatogenesis. Other RNP granules in male germ cells, the reticulated body and CB remnants, are also essential for spermiogenesis. In this study, we disrupted FBXO24, a testis-enriched F-box protein, in mice and found numerous membraneless electron-dense granules accumulated in sperm flagella. *Fbxo24* knockout (KO) mice exhibited malformed flagellar structures, impaired sperm motility, and male infertility, likely due to the accumulation of abnormal granules. The amount and localization of known RNP granule-related proteins were not disrupted in *Fbxo24* KO mice, suggesting that the accumulated granules were distinct from known RNP granules. Further studies revealed that RNAs and two importins, IPO5 and KPNB1, abnormally accumulated in *Fbxo24* KO spermatozoa and that FBXO24 could ubiquitinate IPO5. In addition, IPO5 and KPNB1 were recruited to stress granules, RNP complexes, when cells were treated with oxidative stress or a proteasome inhibitor. These results suggest that FBXO24 is involved in the degradation of IPO5, disruption of which may lead to the accumulation of abnormal RNP granules in sperm flagella.

## eLife assessment

This **important** study reports that FBXO24 is essential for the normal formation and function of the sperm flagellum, motility, and male fertility in mice. The evidence supporting the direct role of this protein in preventing RNP granule formation in the sperm flagellum is **compelling**. This work will be of interest to biomedical researchers who work on testicular biology and male fertility.

## Introduction

Spermatogenesis is a specialized process by which spermatogonia differentiate into spermatocytes, round spermatids, elongating spermatids, and spermatozoa by undergoing meiosis and subsequent spermiogenesis. During spermiogenesis, round spermatids undergo dramatic morphological changes such as nuclear condensation, acrosome formation, flagellum formation, and cytoplasmic removal,

leading to the formation of spermatozoa (*Leblond and Clermont, 1952*). To fertilize eggs, spermatozoa then travel a long distance to reach eggs and pass through the cumulus cell layer and zona pellucida (ZP) that encase eggs. The motility apparatus of spermatozoa is the flagellum which is divided into three parts, a midpiece, a principal piece, and an endpiece, in order from proximal to distal (*Eddy et al., 2003*). The midpiece contains accessory structures called the mitochondrial sheath and outer dense fibers (ODFs) that surround the axoneme, a 9+2 arrangement of microtubules, while the principal piece contains a different accessory structure called the fibrous sheath and ODFs surrounding the axoneme. In contrast, the endpiece contains no accessory structures but the axoneme. Abnormal formation of these flagellar structures could lead to male infertility (*Nsota Mbango et al., 2019*).

Ribonucleoprotein (RNP) granules are membraneless electron-dense structures that assemble through liquid–liquid phase separation and play critical roles in spermatogenesis (*Eddy, 1974*). Among the RNP granules, intermitochondrial cement (IMC) observed in late spermatocytes and the chromatoid body (CB) found in round spermatids are involved in the PIWI-interacting RNA (piRNA) pathway (*Meikar et al., 2011*). After meiosis, IMCs are no longer detectable because of mitochondrial dispersion, and CBs become gradually larger in round spermatids. In elongating spermatids, the CBs containing MIWI, the main effector protein of the piRNA pathway, disappear and two other RNP structures called a reticulated body and a CB remnant that contain TSKS, TSSK1, and TSSK2 appear (*Shang et al., 2010*). Recently, we found that *Tsks* knockout (KO) mice failed to generate reticulated bodies and CB remnants, and were infertile with impaired cytoplasmic removal (*Shimada et al., 2023*), indicating that the formation and disassembly of RNP granules is highly regulated and important for spermiogenesis.

During spermiogenesis, post-translational modifications of proteins are critical since gene transcription is thought to cease with nuclear condensation. Ubiquitination is one of the post-translational modifications in which a small protein called ubiquitin is attached to target proteins. Ubiquitinated proteins are degraded by a ubiquitin-proteasome system (UPS), which is involved in various physiological phenomena in cells, such as cell cycle regulation and signal transduction (*Ciechanover and Schwartz, 1998*; *Mani and Gelmann, 2005*; *Zheng et al., 2016*). Protein ubiquitination is carried out by a cascade of enzymatic reactions, including ubiquitin-activating enzyme (E1), ubiquitin-binding enzyme (E2), and ubiquitin ligase (E3), and target proteins ubiquitinated by E3 ligase are degraded by a large protein complex called the 26S proteasome. Among the approximately 600 E3 ligase genes estimated to be present in humans (*Li et al., 2008*), the mechanism and functions of ubiquitination mediated by the SCF (Skp1-Cul1-F-box protein) complex has been well studied (*Skaar et al., 2013*). The SCF complex consists of several proteins such as SKP1, a cullin protein, an F-box protein, and RBX1, and can ubiquitinate various substrates by using different F-box proteins that recognize distinct substrates. In humans, at least 38 F-box proteins have been identified, but the functions of most F-box proteins remain unclear (*Kipreos and Pagano, 2000*). In this study, we analyzed the function of *Fbxo24* (F-box protein 24), predominantly expressed in testes, and found that FXBO24 is essential for sperm flagellum formation and male fertility.

## Results

### *Fbxo24* is expressed predominantly in male germ cells after meiosis and interacts with SKP1

We performed RT-PCR using cDNAs obtained from multiple mouse tissues and found a specific band for *Fbxo24* in testes (*Figure 1A*). To identify stages in which *Fbxo24* was expressed in spermatogenesis, we performed RT-PCR using postnatal mouse testes over time. *Fbxo24* signals increased after birth, with solid signals observed around postnatal day 21 (*Figure 1B*) when round spermatids appear (*Kluin et al., 1982*). Consistent with the RT-PCR results, single-cell transcriptome data (*Hermann et al., 2018*) indicates that mouse *Fbxo24* and its human orthologue, *FBXO24*, are expressed during the post-meiotic stages (*Figure 1—figure supplement 1A and B*). Pairwise sequence alignment of amino acids sequences revealed that FBXO24 is highly conserved between mice and humans (88% amino acid identity) (*Figure 1—figure supplement 1C*) with the F-box domains located near the N-terminus (*Figure 1—figure supplement 1D*). Because it has been reported that F-box proteins bind to SKP1 via the F-box domain to form the SCF complex (*Bai et al., 1996*; *Schulman et al., 2000*), we investigated the interaction of FBXO24 with SKP1 by co-immunoprecipitation using HEK293T cells. We

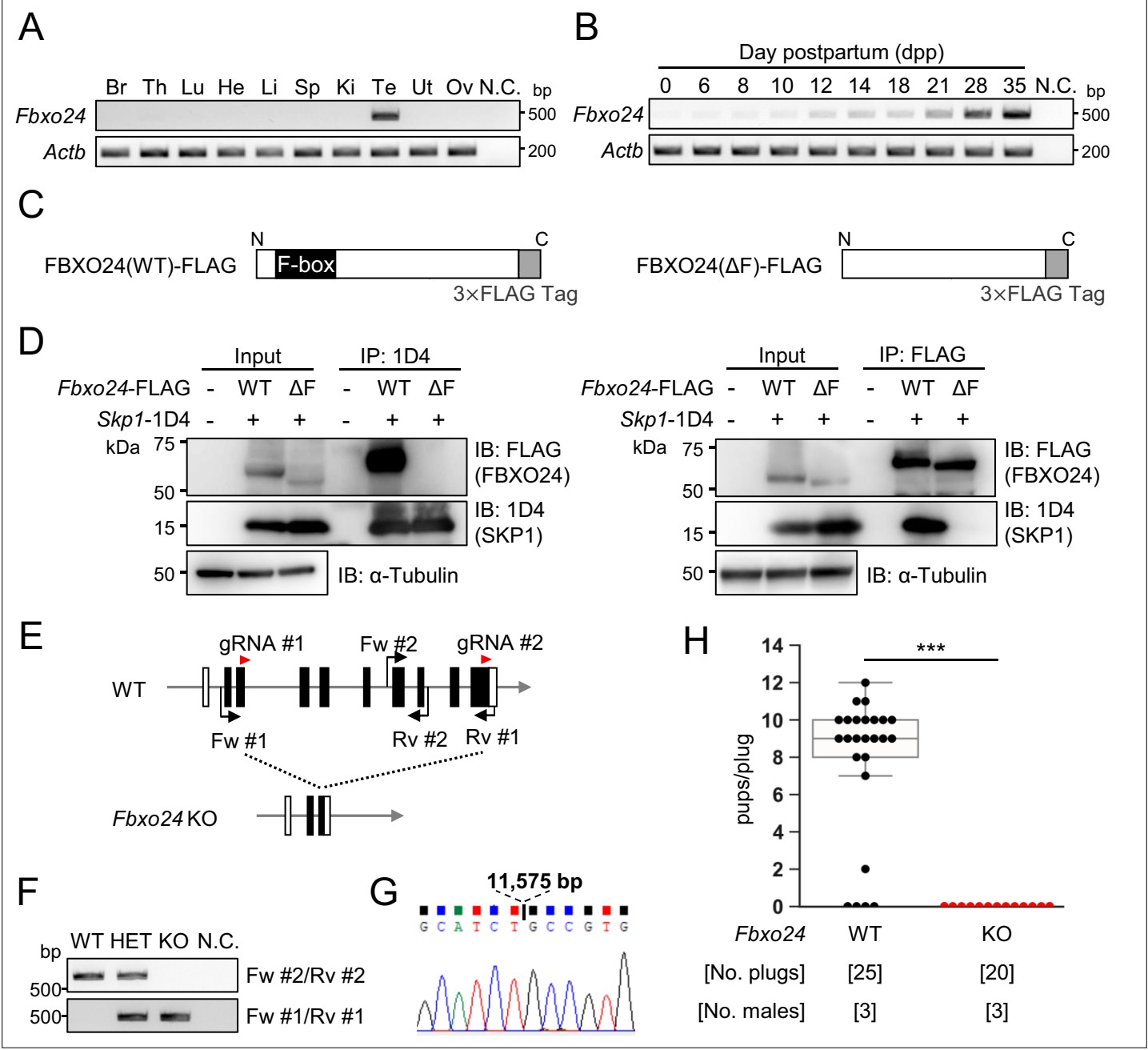

**Figure 1.** *Fbxo24* is predominantly expressed in testes and essential for male fertility. (**A**) RT-PCR of *Fbxo24* in mouse adult tissues. *Fbxo24* is predominantly expressed in the testis. Br: brain, Th: thymus, Lu: lung, He: heart, Li: liver, Sp: spleen, Ki: kidney, Te: testis, Ut: uterus, Ov: ovary, and N.C.: negative control (water). *Actb* was used as a control. (**B**) RT-PCR of *Fbxo24* using RNAs obtained from mouse testes at various postnatal days. *Actb* was used as a control. Water was used as a negative control (N.C.). (**C**) Construction of expression vectors for FBXO24 with (wild-type [WT]) or without (ΔF) the F-box domain. (**D**) *Fbxo24 (WT)-FLAG* or *Fbxo24 (ΔF)-FLAG* was transiently expressed with *Skp1-1D4* in HEK293T cells. Immunoprecipitation (IP) was performed using anti-1D4 antibody or anti-FLAG antibody. FBXO24-FLAG interacts with SKP1-1D4 via the F-box domain. α-Tubulin was used as a loading control. (**E**) Schematic for generating *Fbxo24* knockout (KO) mice using the CRISPR/Cas9 system. White boxes indicate untranslated regions while black boxes indicate protein coding regions. The gRNAs used are shown. Fw and Rv indicate the forward and reverse primer used for genotyping, respectively. (**F**) Genotyping of obtained *Fbxo24* mutant mice. Fw #1-Rv #1 primers for KO allele and Fw #2-Rv #2 primers for WT allele in (E) were used. N.C. indicates negative control (water). (**G**) Amplicons of the PCR product using Fw #1-Rv #1 primers were subjected to direct sequencing and the 11,575 bp deletion was confirmed in the KO allele. (**H**) The number of pups born per plug was counted to assess male fertility. Each WT or KO male was mated with three WT females for 3 months. Error bars are mean ± standard deviation. Statistical significance was assessed with a two-tailed Welch's t-test. ***p<0.001.

*Figure 1 continued on next page*

*Figure 1 continued*

The online version of this article includes the following source data and figure supplement(s) for figure 1:

**Source data 1.** Uncropped and labeled gels for *Figure 1A and B*.

**Source data 2.** Raw unedited gels for *Figure 1A and B*.

**Source data 3.** Uncropped and labeled blots for *Figure 1D*.

**Source data 4.** Raw unedited blots for *Figure 1D*.

**Source data 5.** Uncropped and labeled gels for *Figure 1F*.

**Source data 6.** Raw unedited gels for *Figure 1F*.

**Figure supplement 1.** Characterization of mouse and human FBXO24.

constructed vectors that express FBXO24 tagged with 3×FLAG with (wild-type [WT]) or without (ΔF) the F-box domain (*Figure 1C*). When SKP1-1D4 was immunoprecipitated with an anti-1D4 antibody, WT FBXO24-FLAG was co-immunoprecipitated, whereas ΔF FBXO24-FLAG was not (*Figure 1D*, left). Conversely, when an anti-FLAG antibody was used to immunoprecipitate WT or ΔF FBXO24-FLAG, SKP1-1D4 was co-immunoprecipitated only with WT FBXO24-FLAG (*Figure 1D*, right). These results suggest that FBXO24 could function as a component of the SCF complex in mouse testis, during the post-meiotic stages of spermatogenesis.

## Lack of *Fbxo24* causes male sterility in mice

To examine the role of *Fbxo24* in spermatogenesis, we generated *Fbxo24* KO mice using the CRISPR/Cas9 system. Two gRNAs were designed to delete the majority of the coding region of *Fbxo24* (*Figure 1E*). One hundred fertilized eggs were electroporated, and the resulting 89 two-cell stage embryos were transferred to the oviducts of pseudopregnant ICR mice. One out of 18 born pups contained the large deletion of the coding region and subsequent mating was performed to obtain *Fbxo24* KO mice. We found no overt gross defects in development, behavior, or survival in *Fbxo24* KO mice. We performed genomic PCR using the primers described in *Figure 1E and F* and confirmed that *Fbxo24* KO mice had a deletion of 11,575 bp by Sanger sequencing (*Figure 1G*). *Fbxo24* KO male mice were then mated with three WT females for 3 months to analyze fertility. Although 20 vaginal plugs were detected, no pups were born from the *Fbxo24* KO male mice (*Figure 1H*), suggesting that *Fbxo24* is indispensable for male fertility.

## Disruption of *Fbxo24* results in impaired spermiation and abnormal sperm morphology

We first examined spermatogenesis to determine the cause of male infertility in *Fbxo24* KO mice. No apparent differences in gross testicular morphology (*Figure 2—figure supplement 1A*) or weights (*Figure 2—figure supplement 1B*) were found between controls and *Fbxo24* KO mice. We then performed He-PAS staining of testicular and epididymis sections (*Figure 2A* and *Figure 2—figure supplement 1C*). Although elongating spermatids can be found in *Fbxo24* KO testes, step 16 spermatids were still present in Stage IX seminiferous tubules (*Figure 2A*), indicating that spermiation was impaired in *Fbxo24* KO testes. In contrast, we could not find apparent differences in the cross sections of the cauda epididymis between the two genotypes (*Figure 2—figure supplement 1C*). Next, we observed mature spermatozoa collected from the cauda epididymis. While KO spermatozoa showed comparable head morphology to the controls, KO spermatozoa exhibited abnormal tail structures such as bent or coiled flagella (*Figure 2B and C*). Further analyses of sperm head morphology with immunostaining showed no overt abnormalities in the acrosome or nucleus (*Figure 2—figure supplement 1D*). Since abnormal flagellar morphology could result in decreased motility, we performed computer-assisted sperm analysis (CASA) after 10 and 120 min of incubation in capacitation medium. CASA revealed that the percentages of motile spermatozoa were significantly reduced in *Fbxo24* KO mice compared to the controls (*Figure 2D* and *Figure 2—videos 1 and 2*). Furthermore, all velocity parameters such as average path velocity, straight line velocity, and curvilinear velocity were lower in *Fbxo24* KO spermatozoa (*Figure 2E–G*). These results indicate that FBXO24 is critical in sperm flagellum formation during spermiogenesis.

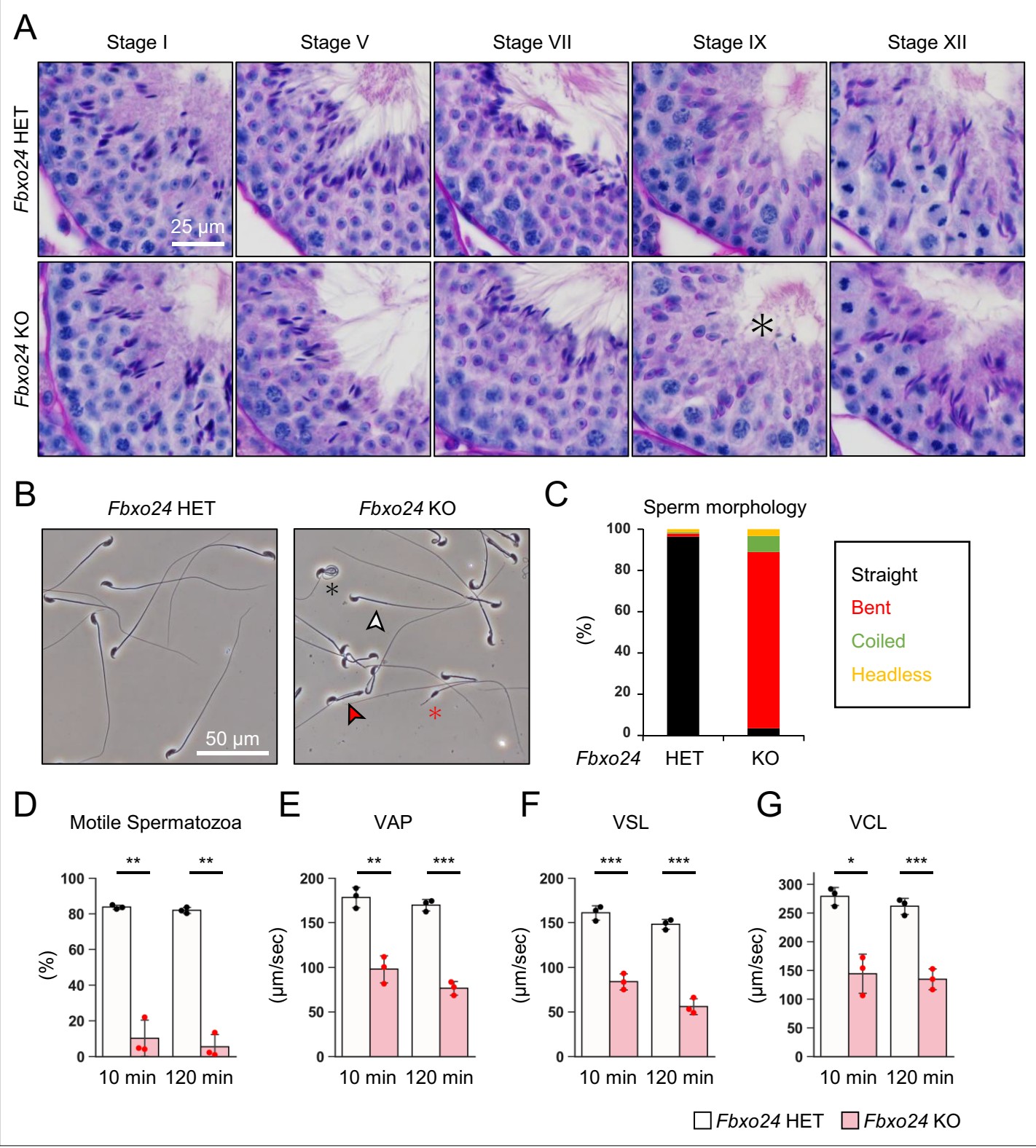

**Figure 2.** FBXO24 is essential for sperm flagellum formation and motility. (**A**) The histology of seminiferous tubules at different stages. An asterisk indicates remaining sperm heads. (**B**) Morphology of mature spermatozoa obtained from cauda epididymis. White arrowhead indicates straight spermatozoa. Red arrowhead indicates bent spermatozoa. Black asterisk indicates coiled spermatozoa. Red asterisk indicates headless spermatozoa. (**C**) Stacked bar graph showing the frequency of sperm morphology classified as straight, bent, coiled, and headless. n = 3 independent experiments. (**D**) Percentages of motile spermatozoa were analyzed at 10 and 120 min after incubation in capacitation medium. (**E**) VAP (average path velocity) was

*Figure 2 continued on next page*

*Figure 2 continued*

analyzed at 10 and 120 min after incubation in capacitation medium. (**F**) VSL (straight line velocity) was analyzed at 10 and 120 min after incubation in capacitation medium. (**G**) VCL (curvilinear velocity) was analyzed at 10 and 120 min after incubation in capacitation medium. Error bars are mean ± standard deviation. Each dot indicates individual mouse. Statistical significance was assessed with a two-tailed Welch's t-test. *p<0.05, **p<0.01, ***p<0.001.

The online version of this article includes the following video and figure supplement(s) for figure 2:

**Figure supplement 1.** Histological analyses of *Fbxo24* knockout (KO) testis and epididymis.

**Figure 2—video 1.** Sperm motility of *Fbxo24* heterozygous mice.

https://elifesciences.org/articles/92794/figures#fig2video1

**Figure 2—video 2.** Sperm motility of *Fbxo24* knockout (KO) mice.

https://elifesciences.org/articles/92794/figures#fig2video2

## *Fbxo24* KO spermatozoa cannot fertilize eggs in vitro and fail to migrate from the uterus into the oviduct

To further evaluate the fertilizing ability of *Fbxo24* KO spermatozoa, we performed in vitro fertilization (IVF). Consistent with male sterility observed in vivo (*Figure 1H*), *Fbxo24* KO spermatozoa could not fertilize eggs in vitro (*Figure 3A*). Furthermore, removing cumulus cells (*Figure 3B*) or both cumulus cells and the ZP (*Figure 3C*) could not rescue the fertilization rates, suggesting that *Fbxo24* KO spermatozoa may have defects in not only sperm motility but also the acrosome reaction, which is a prerequisite for spermatozoa to pass through the ZP and fuse with eggs (*Morohoshi et al., 2023*). Therefore, we crossed *Fbxo24* mutant mice with transgenic (Tg) mice which express EGFP in the acrosome and DsRed2 in the mitochondria (Red Body Green Sperm [RBGS] mice) (*Hasuwa et al., 2010*) and analyzed the acrosome reaction rates. RBGS spermatozoa lose EGFP fluorescent signal when the spermatozoa undergo the acrosome reaction. First, to test whether *Fbxo24* KO spermatozoa were viable, we performed propidium iodide (PI) staining and found that the percentages of dead spermatozoa were significantly higher in *Fbxo24* KO mice (*Figure 3—figure supplement 1A*). We then analyzed the acrosome reaction rates of PI-negative live spermatozoa. While the control spermatozoa underwent the acrosome reaction after 120 min of incubation in capacitation medium and with $Ca^{2+}$ ionophore (A23187) treatment, *Fbxo24* KO spermatozoa rarely underwent the acrosome reaction even after adding A23187 (*Figure 3—figure supplement 1B*). Since the SNARE complex is considered critical for the acrosome reaction (*Hutt et al., 2005*; *Katafuchi et al., 2000*; *Schulz et al., 1997*; *Tomes et al., 2002*), we examined the amounts of SNARE-related proteins in mature spermatozoa using mass spectrometry (MS) and immunoblotting (*Figure 3—figure supplement 1C and D*), but no notable differences were found in SNARE-related proteins between the control and *Fbxo24* KO spermatozoa. Further, no significant differences were found in PLCD4 that has been reported to be important for the acrosome reaction (*Fukami et al., 2001*).

Next, we observed sperm migration in the female reproductive tract using RBGS mice. Although the control spermatozoa migrate through the uterotubal junction (UTJ) 4 hr after mating, *Fbxo24* KO spermatozoa hardly passed through the UTJ (*Figure 3D*). Since the processing of sperm membrane protein, ADAM3, is necessary for sperm migration through the UTJ (*Fujihara et al., 2019*; *Yamaguchi et al., 2006*), we performed immunoblotting analyses and found that ADAM3 was processed correctly even in *Fbxo24* KO spermatozoa (*Figure 3E*). We further performed immunoblotting of LY6K because *Ly6k* KO spermatozoa cannot migrate through the UTJ even with normal ADAM3 processing (*Fujihara et al., 2014*). However, we did not detect significant differences in the amounts of LY6K (*Figure 3—figure supplement 1E*). Considering that the acrosome reaction is not essential for sperm migration in the female reproductive tract (*Morohoshi et al., 2023*) and defects in sperm motility could cause impaired migration through the UTJ (*Chung et al., 2014*; *Fujihara et al., 2018*; *Miyata et al., 2015*; *Shimada et al., 2019*), these results suggest that lower viability and decreased sperm motility can be the cause of abnormal sperm migration and male sterility in vivo.

Next, we performed intracytoplasmic sperm injection (ICSI) to examine whether nuclei of *Fbxo24* KO spermatozoa have the potential to generate the next generation. We injected *Fbxo24* KO sperm heads into 91 WT oocytes, and obtained 48 two-cell stage embryos (*Figure 3F*). By transplanting the two-cell embryos into the oviduct of pseudopregnant ICR females, we obtained four pups, which were

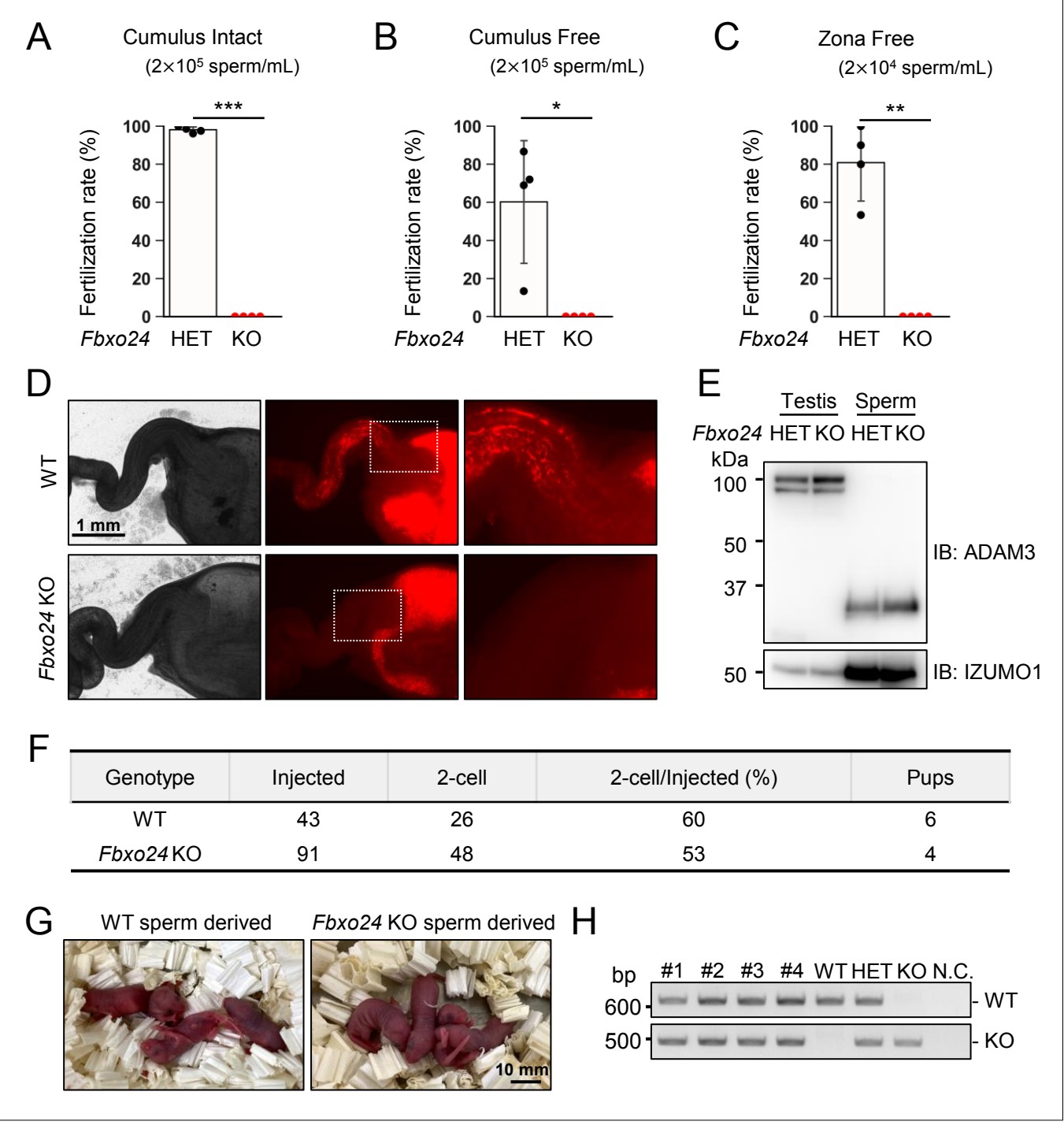

**Figure 3.** In vitro fertilizing ability and in vivo sperm migration. (**A**) The fertilizing ability of spermatozoa was analyzed in vitro using cumulus-intact oocytes. (**B**) The fertilizing ability of spermatozoa was analyzed in vitro using cumulus-free oocytes. (**C**) The fertilizing ability of spermatozoa was analyzed in vitro using zona-free oocytes. (**D**) Uterus and oviducts of wild-type (WT) females mated with WT or *Fbxo24* knockout (KO) males carrying Red Body Green Sperm (RBGS) transgene. Female reproductive tracts were dissected 4 hr after confirming a plug. Right figures are magnified images of the boxes indicated in the middle panels. (**E**) Immunoblotting of ADAM3 using testis and mature spermatozoa of *Fbxo24* heterozygous or KO mice. (**F**) Intracytoplasmic sperm injection (ICSI) experiment. The number of two-cell stage embryos and pups developed from WT oocytes injected with WT or *Fbxo24* KO spermatozoa. (**G**) Pups derived from WT or *Fbxo24* KO spermatozoa. (**H**) Genotyping of pups obtained from *Fbxo24* KO spermatozoa. N.C. indicates negative control (water). Error bars are mean ± standard deviation. Each dot indicates individual mouse. Statistical significance was assessed with a two-tailed Welch's t-test. *p<0.05, **p<0.01, ***p<0.001.

*Figure 3 continued on next page*

*Figure 3 continued*

The online version of this article includes the following source data and figure supplement(s) for figure 3:

**Source data 1.** Uncropped and labeled blots for *Figure 3E*.

**Source data 2.** Raw unedited blots for *Figure 3E*.

**Source data 3.** Uncropped and labeled gels for *Figure 3H*.

**Source data 4.** Raw unedited gels for *Figure 3H*.

**Figure supplement 1.** Lack of *Fbxo24* impairs sperm viability and the acrosome reaction.

**Figure supplement 1—source data 1.** Uncropped and labeled blots for *Figure 3—figure supplement 1D and E*.

**Figure supplement 1—source data 2.** Raw unedited blots for *Figure 3—figure supplement 1D and E*.

confirmed to have the heterozygous mutation by PCR (*Figure 3G and H*). These results indicate that ICSI can rescue male sterility of *Fbxo24* KO mice.

## FBXO24 is required for the sperm midpiece formation

We analyzed mitochondria localization in *Fbxo24* KO spermatozoa because abnormal mitochondrial sheath structures were found in spermatozoa with bent tails, such as in *Armc12* KO, *Tbc1d21* KO, and *Gk2* KO mice (*Shimada et al., 2021*; *Shimada et al., 2019*). We observed midpieces using RBGS Tg mice and revealed that mitochondria were disorganized in *Fbxo24* KO spermatozoa (*Figure 4A*). We also observed SEPT4, a component of the annulus (*Kissel et al., 2005*), and found that SEPT4 localized to the proper region (*Figure 4—figure supplement 1A*).

We then used a scanning electron microscope (SEM) to observe the formation of mitochondrial sheaths. During the early step of mitochondrial sheath formation, mitochondria wrap around the axoneme. In both control and *Fbxo24* KO spermatids, spherical mitochondria were aligned correctly at this step (*Figure 4B*, left panels). In the next step, mitochondria became crescent-shaped and were interlocked in the control spermatids, whereas irregular interlocking was observed in *Fbxo24* KO spermatids (*Figure 4B*, middle panels). Subsequently, elongating mitochondria remain irregularly arranged in *Fbxo24* KO spermatids (*Figure 4B*, right panels).

To further investigate the morphological abnormality of the midpiece in *Fbxo24* KO mice, we observed the ultrastructure of the cauda epididymal spermatozoa using transmission electron microscopy (TEM). Consistent with the SEM results, we found that mitochondria were irregularly arranged in *Fbxo24* KO spermatozoa (*Figure 4C and D*). The frequency of abnormal mitochondria was significantly higher in *Fbxo24* KO spermatozoa (*Figure 4E*). In addition, the axoneme and ODF were disrupted in both midpieces and principal pieces of *Fbxo24* KO spermatozoa (*Figure 4—figure supplement 1B–E*). Furthermore, we frequently observed numerous membraneless electron-dense granules in the midpieces of *Fbxo24* KO spermatozoa (*Figure 4C, D, and F*). We found these granules even in the axoneme (*Figure 4C*). Despite the low frequency, we also found membraneless electron-dense granules in the principal pieces of *Fbxo24* KO spermatozoa (*Figure 4—figure supplement 1F and G*).

Because membraneless electron-dense granules accumulated in *Fbxo24* KO spermatozoa, we examined RNP granule-related proteins in *Fbxo24* KO mice. We performed immunoblotting analyses for ADAD1, ADAD2, MILI, MIWI, RNF-17, YTHDC2, TSKS, and TSSK1, which localized in germ-cell RNP granules and are essential for spermatogenesis (*Lu et al., 2023*; *Meikar et al., 2011*; *Pan et al., 2005*; *Shang et al., 2010*; *Shimada et al., 2023*; *Snyder et al., 2020*), but we did not see any differences in the amounts of these proteins between *Fbxo24* heterozygous and KO testes (*Figure 5—figure supplement 1A*). Further, we found no abnormalities in the localization of MIWI in *Fbxo24* KO testes (*Figure 5—figure supplement 1B*). MIWI was localized in the IMCs of late spermatocytes and CBs of round spermatids and disappeared in elongating spermatids (*Figure 5—figure supplement 1B*). Further, no abnormalities were found in the immunostaining of TSKS, which localized in the reticulated body and CB remnant of elongating spermatids (*Figure 5—figure supplement 1C*).

## IPO5 and KPNB1 amounts increase in *Fbxo24* KO flagella

Since previous studies show that loss of F-box protein causes the accumulation of substrates (*Nakayama et al., 2000*; *Tsunematsu et al., 2004*), we investigated whether certain proteins accumulated in *Fbxo24* KO spermatozoa. MS analyses revealed that the amounts of several proteins significantly

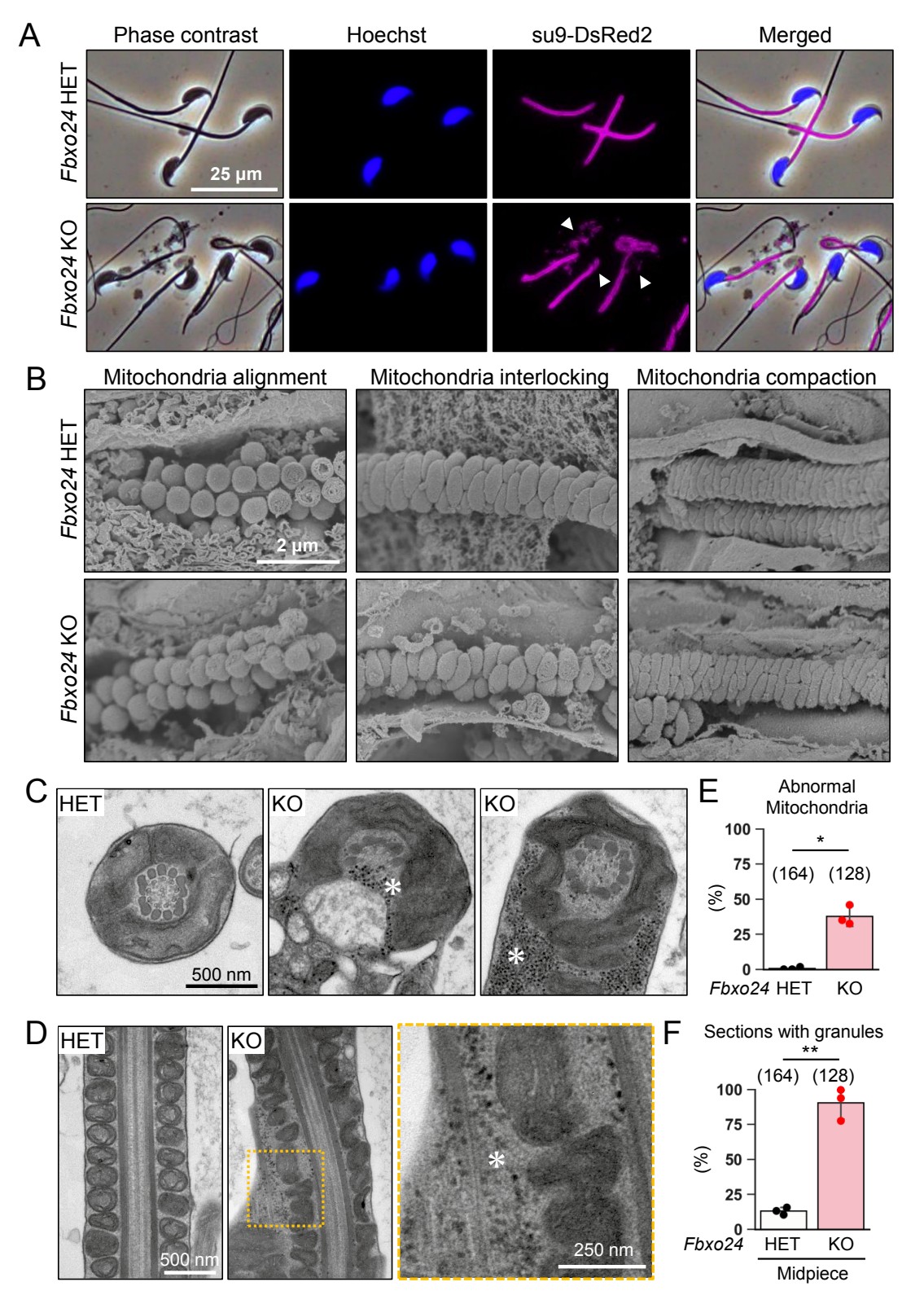

**Figure 4.** Numerous membraneless electron-dense granules were found in *Fbxo24* knockout (KO) spermatozoa. (**A**) Epididymal spermatozoa of Red Body Green Sperm (RBGS) Tg mice were stained with Hoechst 33342 (nuclei). Mitochondria were labeled with su9-DsRed2. White arrowheads indicate disorganized mitochondria. (**B**) Sperm mitochondrial sheath formation during spermiogenesis was observed by scanning electron microscopy (SEM). (**C**) Cross sections of spermatozoa in the cauda epididymis. Asterisks indicate electron-dense granules. (**D**) Longitudinal sections of spermatozoa in the

*Figure 4 continued on next page*

Figure 4 continued

cauda epididymis. An asterisk indicates electron-dense granules. (**E**) Percentages of morphologically abnormal mitochondria observed with transmission electron microscopy (TEM). The number of flagellar sections analyzed is shown above each bar. (**F**) Percentages of electron-dense granules observed in the midpiece cross sections with TEM. The number of flagellar sections analyzed is shown above each bar. Error bars are mean ± standard deviation. Each dot indicates individual mouse. Statistical significance was assessed with a two-tailed Welch's t-test. *p<0.05, **p<0.01.

The online version of this article includes the following figure supplement(s) for figure 4:

**Figure supplement 1.** Sperm flagellum ultrastructure was disorganized in *Fbxo24* knockout (KO) mice.

increased in *Fbxo24* KO mature spermatozoa (*Figure 5A*). Among the significantly increased proteins, we focused on IPO5 (importin 5, also known as KPNB3 and RanBP5) and KPNB1 (karyopherin subunit beta 1, also known as importin β1) because the fold changes (KO/WT) are the highest in these two proteins. Both IPO5 and KPNB1 are members of the karyopherin-β (KPNB) family of nuclear transport receptors (NTRs), which play a crucial role in nucleocytoplasmic transport (*Chook and Blobel, 2001*; *Chook and Süel, 2011*; *Conti and Izaurralde, 2001*; *Kimura and Imamoto, 2014*). In rodent testes, KPNB1 localized to the cytoplasm of spermatogonia, spermatocytes, and Sertoli cells, and IPO5 localized to the cytoplasm of elongating spermatids (*Loveland et al., 2015*; *Loveland et al., 2006*). Immunoblotting analyses confirmed that the amounts of IPO5 and KPNB1 increased in *Fbxo24* KO mature spermatozoa (*Figure 5B*). In contrast, the amounts of KPNA2 (importin α1), a member of karyopherin-α (KPNA) family of NTRs which localized to the cytoplasm and nucleus of spermatocytes and the cytoplasm of elongating spermatids (*Miyamoto et al., 2013*), were comparable between the control and *Fbxo24* KO spermatozoa. Since the anti-KPNB1 antibody did not work for immunostaining, we performed immunostaining of mature spermatozoa using an anti-IPO5 antibody. We detected IPO5 in the *Fbxo24* KO flagella but not in control (*Figure 5C*).

## FBXO24 could interact with IPO5

Next, we investigated the proteins that interacted with FBXO24. Because we could not obtain anti-FBXO24 antibodies that worked for immunoprecipitation, we generated Tg mice expressing 3×FLAG-tagged *Fbxo24* under a testis-specific *Prm1* promoter (*Figure 5—figure supplement 2A and B*). This transgene could not rescue *Fbxo24* KO sperm morphology (*Figure 5—figure supplement 2C*). By looking for long non-coding RNAs, we found *Gm36266* that is located around Exon 9 and 10 of *Fbxo24*, which is deleted in *Fbxo24* KO mice (*Figure 1E*); however, a partial deletion of Exon 2 and 3 of *Fbxo24* showed the same defects in sperm morphology (*Figure 5—figure supplement 2D–G*) as mice lacking Exon 3–10 (*Figure 2B*), suggesting that *Fbxo24*, not *Gm36266*, is responsible for the phenotypes observed in this study. Abnormal sperm morphology was not rescued by the transgene, likely due to lower expression of FBXO24 (*Figure 5D* and *Figure 5—figure supplement 2B*) and/or FLAG-tag interfering with FBXO24 function. Using Tg mice, we could immunoprecipitate FBXO24-FLAG with an anti-FLAG antibody, and subsequent MS analyses detected not only FBXO24 but also IPO5 with the highest quantitative values (*Figure 5—figure supplement 2H*), indicating that these proteins interact in vivo. We also found SKP1 in this analysis, consistent with the in vitro study (*Figure 1D*). We confirmed that IPO5 co-immunoprecipitated with FBXO24-FLAG in mouse testes using immunoblotting analysis (*Figure 5D*). Further, we transiently expressed 3×FLAG-tagged *Fbxo24* in HEK293T cells, performed immunoprecipitation analysis using an anti-FLAG antibody, and found that FBXO24 could interact with endogenous IPO5 but not with endogenous KPNB1 (*Figure 5E*). To examine if FBXO24 can ubiquitinate IPO5, we performed a ubiquitination assay using HEK293T cells and revealed that ubiquitination of IPO5 was upregulated when cells were transfected with WT FBXO24, but not with ΔF FBXO24 that lacks the F-box domain (*Figure 5F*). These results suggest that FBXO24 recognizes and ubiquitinates IPO5 for subsequent protein degradation.

## IPO5 is recruited into RNP granules under stress conditions

The amounts of known RNP granule-related proteins were not upregulated in *Fbxo24* KO testes (*Figure 5—figure supplement 1A*) and spermatozoa (*Figure 5A*). In contrast, the total amount of RNAs was significantly increased in *Fbxo24* KO spermatozoa (*Figure 6A and B*) although the contamination of somatic cells or immature spermatogenic cells were rarely found (*Figure 6—figure supplement 1*). Previous studies showed that various stresses, such as heat shock, arsenite treatment, or proteasome inhibition, trigger the formation of cytoplasmic stress granules (SGs), membraneless

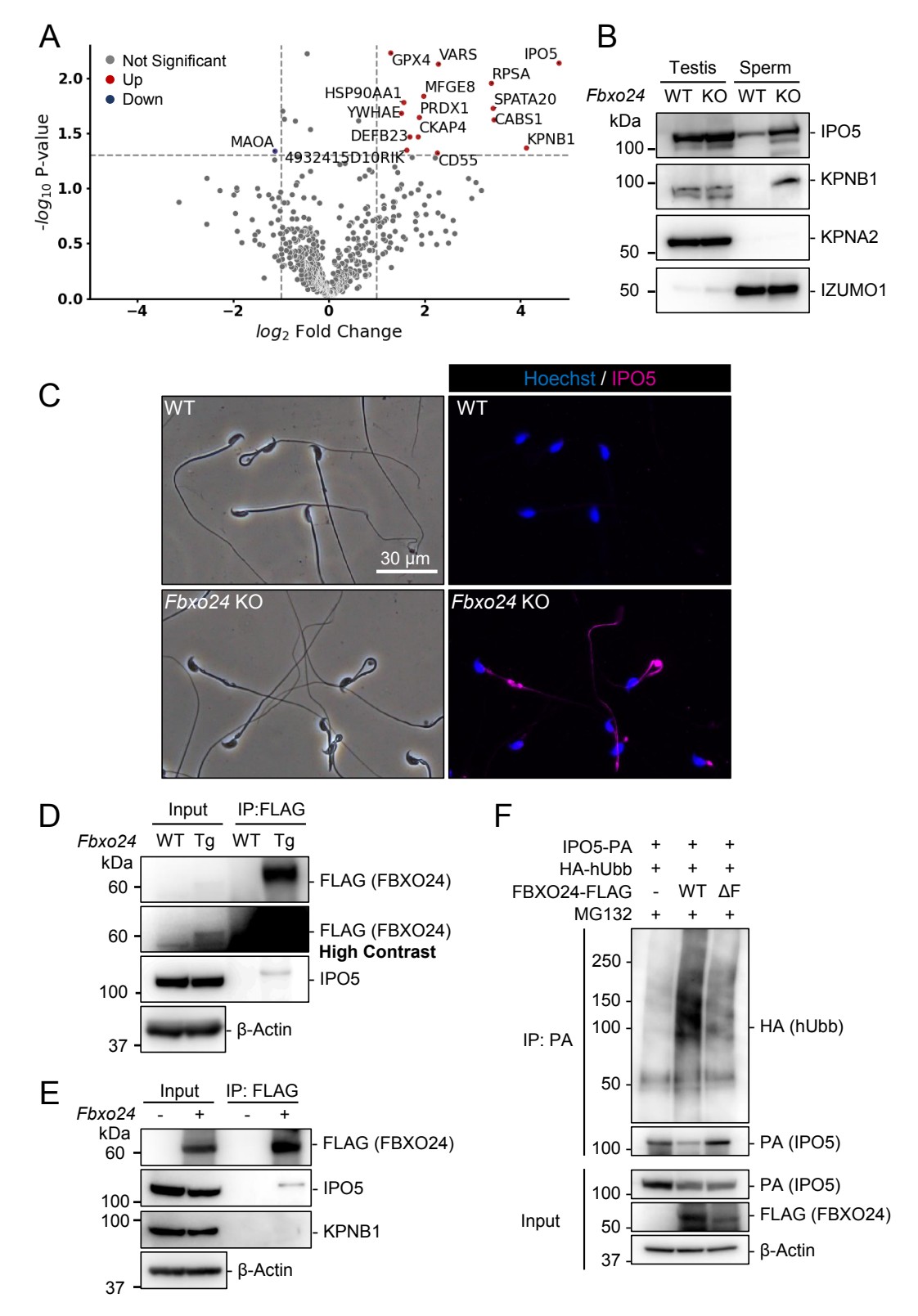

**Figure 5.** IPO5 and KPNB1 accumulate in *Fbxo24* knockout (KO) spermatozoa. (**A**) Mass spectrometry analyses of mature spermatozoa. Significantly upregulated proteins are shown with red dots whereas significantly downregulated proteins are shown with blue dots. (**B**) Immunoblotting analysis was performed using proteins extracted from testes or mature spermatozoa. IPO5 was detected using rabbit anti-IPO5 polyclonal antibody. KPNA2 and IZUMO1 were detected as negative control and loading control, respectively. (**C**) Spermatozoa obtained from the cauda epididymis were stained for

*Figure 5 continued on next page*

*Figure 5 continued*

IPO5 (magenta) using rabbit anti-IPO5 polyclonal antibody. Nuclei were stained with Hoechst 33342 (blue). (**D**) Immunoprecipitation (IP) of FBXO24-FLAG from *Fbxo24-FLAG* Tg testes. FBXO24 could interact with IPO5. IPO5 was detected using mouse anti-IPO5 monoclonal antibody. IPO5 band is slightly larger after IP likely due to different protein composition in the sample. β-Actin was used as a loading control. A picture with high contrast was shown for the input FLAG band. (**E**) IP of FBXO24-FLAG using HEK293T cells. FBXO24 could interact with IPO5 but not with KPNB1. IPO5 was detected using mouse anti-IPO5 monoclonal antibody. β-Actin was used as a loading control. (**F**) A ubiquitination assay of IPO5 using HEK293T cells. IPO5-PA was immunoprecipitated and the level of ubiquitination was analyzed using anti-HA antibody. β-Actin was used as a loading control.

The online version of this article includes the following source data and figure supplement(s) for figure 5:

**Source data 1.** Uncropped and labeled blots for *Figure 5B, D, E, and F*.

**Source data 2.** Raw unedited blots for *Figure 5B, D, E, and F*.

**Figure supplement 1.** No clear differences were found in the amount and localization of ribonucleoprotein (RNP) granule-related proteins between control and *Fbxo24* knockout (KO) testes.

**Figure supplement 1—source data 1.** Uncropped and labeled blots for *Figure 5—figure supplement 1A*.

**Figure supplement 1—source data 2.** Raw unedited blots for *Figure 5—figure supplement 1A*.

**Figure supplement 2.** Generation and analyses of *Fbxo24–3×FLAG* Tg mice.

**Figure supplement 2—source data 1.** Uncropped and labeled blots for *Figure 5—figure supplement 2B*.

**Figure supplement 2—source data 2.** Raw unedited blots for *Figure 5—figure supplement 2B*.

**Figure supplement 2—source data 3.** Uncropped and labeled gels for *Figure 5—figure supplement 2E*.

**Figure supplement 2—source data 4.** Raw unedited gels for *Figure 5—figure supplement 2E*.

granules composed of RNPs. Further, it has been shown that SGs could contain KPNB1 (*Chang and Tarn, 2009*; *Mahboubi et al., 2013*). Increased amounts of KPNB1 and RNAs suggest that RNP granules may be formed in *Fbxo24* KO spermatozoa. We analyzed whether IPO5 can localize to SGs under stress conditions to explore this possibility. We examined the subcellular localization of endogenous IPO5 and KPNB1 in COS7 cells treated with arsenite, an oxidative stress inducer. Immunostaining analyses revealed that IPO5 was predominantly localized to the cytoplasm without the stress inducer (*Figure 6C*); however, when cells were treated with arsenite, IPO5 was detected not only in the cytoplasm but also in cytoplasmic granules that were colocalized with KPNB1 (*Figure 6C*). We then analyzed if the stress of proteasome inhibition could cause a similar response. We performed immunostaining of IPO5 and KPNB1 in COS7 cells using a proteasome inhibitor, MG132. Consistent with arsenite, MG132 caused the accumulation of cytoplasmic granules that contained IPO5 and KPNB1 (*Figure 6D*). These results support the idea that RNP granules that contain KPNB1 and IPO5 are formed in *Fbxo24* KO spermatozoa.

## Discussion

Spermiogenesis is the late stage of spermatogenesis in which round spermatids differentiate into spermatozoa accompanied by drastic morphological changes such as flagellum formation, chromatin remodeling, and cytoplasmic removal. Protein ubiquitination of target proteins for degradation by the UPS is crucial to maintain functional spermatogenesis (*Richburg et al., 2014*). In this study, we focused on testis-enriched F-box protein FBXO24. We confirmed that *Fbxo24* was predominantly expressed in the testis, and its expression level increased in the late stage of spermatogenesis. The analyses using cultured cells and Tg testes suggested that FBXO24 could interact with SKP1 to form the SCF complex. Further, we deleted *Fxbo24* using the CRISPR/Cas9 system and found that *Fbxo24* KO mice exhibited abnormal flagellar structures and male infertility. *Fxbo24* KO spermatozoa failed to fertilize eggs even with IVF, while ICSI gave offspring.

More detailed observation of the midpiece of *Fbxo24* KO spermatozoa using electron microscopy revealed abnormalities in mitochondria coiling, which occurred in the late stages of spermatogenesis and was consistent with the *Fbxo24* expression timing. Furthermore, *Fbxo24* KO spermatozoa showed markedly reduced motility. These results suggest that the primary cause of infertility is impaired sperm migration in the female reproductive tract due to reduced sperm motility. In addition to the abnormal flagellar morphology, *Fbxo24* KO spermatozoa could not undergo the acrosome reaction even after the A23187 treatment. We analyzed the amounts of SNARE-related proteins and PLCD4; however, no overt differences were found between the control and *Fbxo24* KO spermatozoa. Recently, we

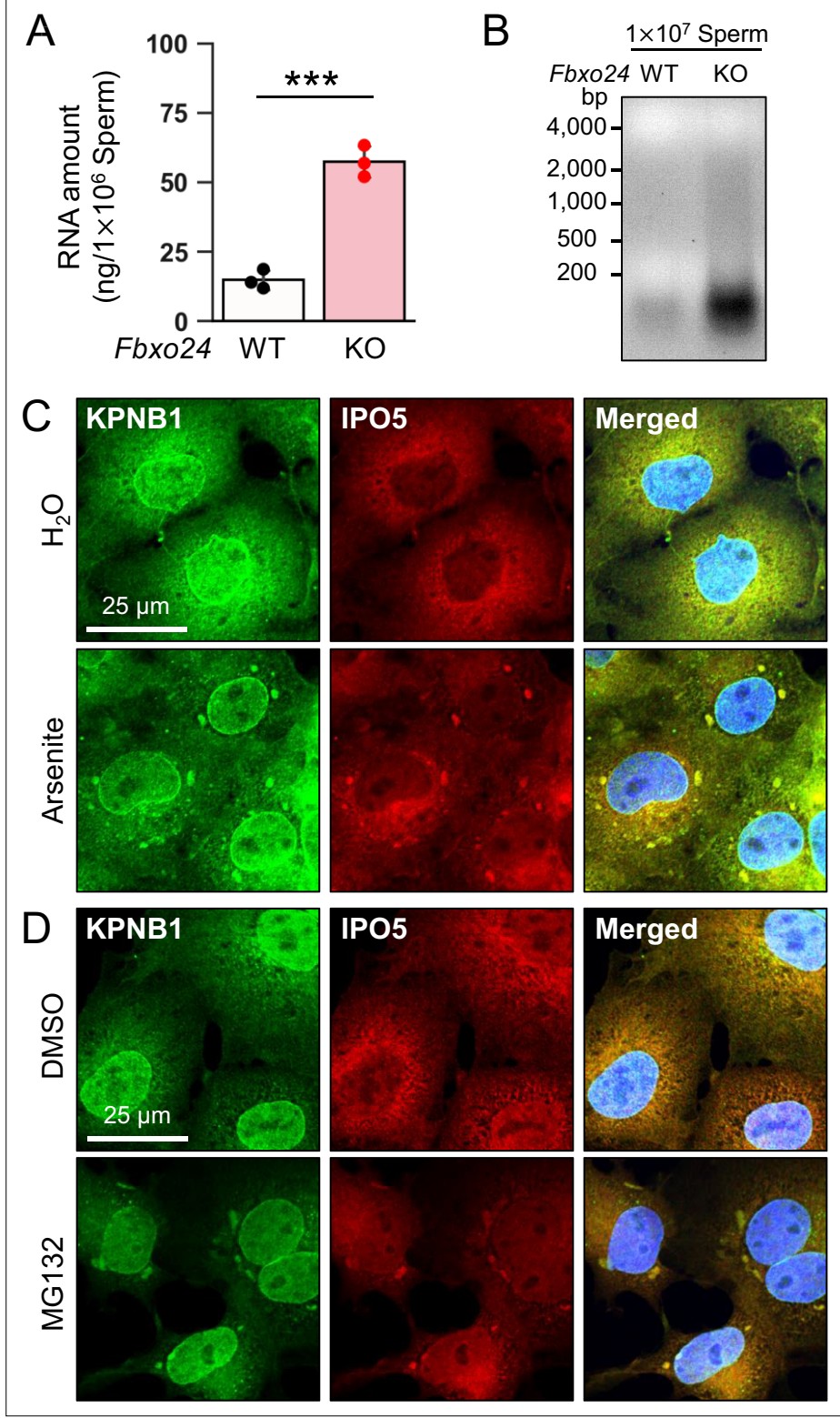

**Figure 6.** KPNB1 and IPO5 are recruited to stress granules (SGs). (**A**) Total RNA amounts in spermatozoa were measured by ultraviolet absorption. (**B**) Electrophoresis of RNA extracted from mature spermatozoa. (**C**) KPNB1 and IPO5 were localized to SGs under exposure to oxidative stress. COS7 cells were treated with water (upper row) or arsenite (lower row). Nuclei were stained with Hoechst 33342 (blue). (**D**) KPNB1 and IPO5 were localized to SGs under exposure to a proteasome inhibitor. COS7 cells were treated with DMSO (upper row) or MG132 (lower

*Figure 6 continued on next page*

*Figure 6 continued*

row). Nuclei were stained with Hoechst 33342 (blue). Error bars are mean ± standard deviation. Each dot indicates individual mouse. Statistical significance was assessed with a two-tailed Welch's t-test. ***p<0.001.

The online version of this article includes the following source data and figure supplement(s) for figure 6:

**Source data 1.** Uncropped and labeled gels for *Figure 6B*.

**Source data 2.** Raw unedited gels for *Figure 6B*.

**Figure supplement 1.** Spermatogenic cells obtained from cauda epididymis.

---

found that FER1L5 is important for $Ca^{2+}$-dependent acrosome reaction (*Morohoshi et al., 2023*), but no antibodies are available to analyze the amount of FER1L5. It is possible that FER1L5 or unknown proteins involved in the acrosome reaction deteriorated in *Fbxo24* KO spermatozoa. Alternatively, it is possible that acrosome reaction failure may be caused by impaired sperm capacitation associated with abnormal flagellar structures.

The TEM analyses observed numerous membraneless electron-dense granules in *Fbxo24* KO sperm flagella. Although the irregular arrangement of sperm mitochondria has been reported in KO mice of some mitochondria-associated proteins (*Shimada et al., 2021*; *Shimada et al., 2019*), numerous granules have not been observed in these KO spermatozoa. Proteomic analyses revealed that IPO5 and KPNB1 accumulated in *Fbxo24* KO spermatozoa without significant elevation of mitochondria-associated proteins. The lack of accumulation of KPNA2, which is part of the importin family, also suggests that there is selectivity in the accumulated proteins. Abnormal accumulation of proteins, including IPO5 and KPNB1, may lead to morphological abnormalities in sperm flagella since the SCF complex is critical for protein degradation and consequently regulate various cellular processes (*Skaar et al., 2013*). FBXO24 likely plays a role in catalyzing protein degradation during spermiogenesis. Because we found that FBXO24 could interact with IPO5 in testes and catalyze ubiquitination of IPO5, FBXO24 may degrade IPO5 in the testis. Recently, mice lacking both PSME4 and ECPAS, proteasome-associated proteins, were shown to exhibit impaired male fertility with disorganized mitochondrial sheath, but numerous membraneless granules were not observed (*Sato et al., 2023*), suggesting that the accumulation of the granules is specific to *Fbxo24* deletion rather than simply caused by proteasome defects.

In addition to proteins, the total RNA amount increased in *Fbxo24* KO spermatozoa, suggesting that the abnormal granules observed in *Fbxo24* KO flagella may consist of RNPs. In germ cells, RNP granules have been well studied and shown to be essential for spermatogeneses, such as IMCs, CBs, reticulated bodies, and CB remnants (*Chuma et al., 2006*; *Meikar et al., 2011*; *Shimada et al., 2023*). RNP granules are membraneless electron-dense structures that are similar to the abnormal granules in *Fbxo24* KO; however, the localization of MIWI, an IMC and CB marker, and TSKS, a reticulated body and CB remnant marker, was not disrupted in the KO. These results suggest that electron-dense granules accumulated in *Fbxo24* KO spermatozoa are different from well-studied RNP granules. SGs are also membraneless electron-dense granules that contain RNPs, which assemble through liquid–liquid phase separation in response to cellular stresses (*Buchan and Parker, 2009*; *Protter and Parker, 2016*; *Taylor et al., 2016*). While transient SGs formed in response to oxidative stress or heat shock play roles in translational arrest to ensure cell survival, chronic SGs are relevant to neurodegenerative diseases (*Marcelo et al., 2021*; *Reineke and Neilson, 2019*). It has been shown that, under arsenite-induced oxidative stress, KPNB1 relocates to SGs (*Chang and Tarn, 2009*; *Mahboubi et al., 2013*). Because we found that not only KPNB1 but also IPO5 can be recruited to SGs after arsenite or proteasome inhibitor treatment, electron-dense granules observed in *Fbxo24* KO spermatozoa may be RNP granules consisting of IPO5 and KPNB1. Our results suggest that FBXO24 is responsible for degrading proteins such as IPO5, which may prevent the accumulation of abnormal RNP granules. While this paper was under review, another group reported analyses of *Fbxo24* KO mice that exhibit not only disorganized mitochondrial coiling but also abnormal head morphology (*Li et al., 2024*). The differences from our study may be due to different mouse genetic backgrounds. The other group showed that FBXO24 mediates the degradation of MIWI via K48-linked polyubiquitination (*Li et al., 2024*). FBXO24 may be involved in the degradation of multiple proteins including IPO5 to support proper spermiogenesis.

In conclusion, we reveal that numerous RNP granules accumulated in sperm flagella when FBXO24, which is related to the UPS, was depleted. Small numbers of the RNP granules were found even in the control, suggesting that the ability to generate the granules is present in WT testes but may be suppressed by FBXO24. FBXO24 can be a key molecule for understanding the formation and function of the aberrant RNP granules in male fertility. Because amino acid sequences and expression patterns of FBXO24 are conserved, FBXO24 may also play similar roles in human testes. Our research may lead to development of new approaches to infertility treatment and nonhormonal male contraceptives.

## Materials and methods

### Animals
All animal experiments were approved by the Animal Care and Use Committee of the Research Institute for Microbial Diseases, Osaka University (#Biken-AP-H30-01 and #Biken-AP-R03-01). Mice were purchased from CLEA Japan (Tokyo, Japan) or Japan SLC (Shizuoka, Japan). WT or *Fbxo24* heterozygous (HET) mice were used as controls. All gene-modified mice generated in this study will be made available through either the RIKEN BioResource Research Center or the Center for Animal Resources and Development (CARD), Kumamoto University.

### Isolation of RNA and RT-PCR
Adult mouse multi-tissues and mouse testes at different ages were obtained from C57BL/6N mice. RNA samples were isolated and purified using TRIzol (Thermo Fisher Scientific, Waltham, MA, USA). RNA was reverse transcribed to cDNA using SuperScript IV First-Strand Synthesis System (Thermo Fisher Scientific) using an oligo (dT) primer. PCR was performed using the KOD Fx Neo DNA Polymerase (Toyobo, Tokyo, Japan). Primers used in this study are listed in *Supplementary file 1*.

### Transfection of HEK293T cells and induction of stress
HEK293T cells (*Tiscornia et al., 2006*) and COS7 cells (#RCB0539, RIKEN BioResource Research Center) were maintained in Dulbecco's Modified Eagle Medium supplemented with 10% fetal bovine serum (Sigma-Aldrich, St. Louis, MO, USA) and 1% Gibco penicillin/streptomycin (Thermo Fisher Scientific) at 37°C under 5% $CO_2$. For transfection, HEK293T cells were transiently transfected with the plasmid DNA using the calcium phosphate transfection method (*Tiscornia et al., 2006*), and cultured for 24 hr before harvesting. For oxidative stress, COS7 cells were treated with 0.5 mM sodium arsenite in medium for 30 min and controls were incubated with water in medium. For proteasome inhibition, COS7 cells were treated with 10 μM sodium MG132 (Sigma-Aldrich) in medium for 3 hr and controls were incubated with DMSO in medium.

### Generation of *Fbxo24* KO mice
*Fbxo24* KO mice were generated using CRISPR/Cas9. Three crRNAs, 5'- TGTGGAGGCGCATCTG TCGA –3', 5'- TCCTGAAGGAAGTCGAGCCG –3' and 5'- TCAGTTGTTCCCCCCAGAGC –3' were designed using the online source CRISPRdirect (*Naito et al., 2015*) and annealed to SygRNA Cas9 Synthetic tracrRNA (#TRACRRNA05N-5NMOL, Sigma-Aldrich). The gRNAs were mixed with TrueCut Cas9 Protein v2 (A36498, Thermo Fisher Scientific) and incubated at 37°C for 5 min to form CRISPR/Cas9 complexes. The complexes were introduced into fertilized eggs which were from superovulated WT B6D2F1 females mated with B6D2F1 males. Electroporation was performed using the super electroporator NEPA21 (NEPA GENE, Chiba, Japan) (poring pulse, voltage: 225 V, pulse width: 2 ms, pulse interval: 50 ms, number of pulses: +4, and attenuation: 10%; transfer pulse, voltage: 20 V, pulse width: 50 ms, pulse interval: 50 ms, number of pulses: ±5, and attenuation: 40%). The treated eggs were developed into two-cell-stage embryos by cultivating in KSOM medium (*Ho et al., 1995*) and transplanted into pseudopregnant ICR females. The obtained pups were genotyped by PCR to detect the KO and/or WT allele and then subjected to Sanger sequencing to verify the deleted sequence.

### Fertility test
For the in vivo fertility analysis, sexually mature KO male mice or WT male mice were caged with three 8-week-old B6D2F1 female mice for 3 months and plugs were checked every morning. The number of pups was counted on the day of birth. For the in vitro fertility assay, IVF analysis was performed as

previously described (*Morohoshi et al., 2021*) with some minor changes. For ZP-free oocytes, sperm insemination was performed at a final density of $2 \times 10^4$ spermatozoa/ml.

## Histological analysis

PAS staining of sections were performed as previously described (*Morohoshi et al., 2020*). Testes or cauda epididymis were fixed at 4°C in Bouin's solution (Polysciences, Inc, Warrington, PA, USA) and were processed for paraffin embedding. Paraffin sections were cut at a thickness of 5 μm using an HM325 microtome (Microm, Walldorf, Germany). After rehydrating the sections, they were stained with 1% periodic acid (Nacalai Tesque, Kyoto, Japan) and Schiff's reagent (FUJIFILM WakoPure Chemical, Osaka, Japan) for 20 min each at room temperature. The sections were then counterstained with Mayer's hematoxylin solution (FUJIFILM WakoPure Chemical). The sections were observed with an Olympus BX-53 microscope (Tokyo, Japan).

## Morphological and motility analysis of spermatozoa

Spermatozoa extracted from cauda epididymis were suspended in TYH medium (*Muro et al., 2016*). After 10 min incubation, spermatozoa were collected to observe morphology. Sperm motility was analyzed as previously described (*Miyata et al., 2021*; *Miyata et al., 2020*). The motility of more than 200 spermatozoa was measured after incubation at 10 and 120 min in TYH medium using CEROS II (software version 1.4; Hamilton Thorne Biosciences, Beverly, MA, USA).

## Observation of sperm flagellum ultrastructure using TEM

Cauda epididymis specimens were prepared as previously described (*Shimada et al., 2019*). The prepared samples were observed using a JEM-1400 plus electron microscope (JEOL, Tokyo, Japan) at 80 kV with a CCD Veleta 2K × 2K camera (Olympus).

## Observation of sperm mitochondria during spermatogenesis using SEM

Testes specimens were prepared as previously described (*Shimada et al., 2019*). The prepared samples were observed using an S-4800 field emission SEM (Hitachi, Tokyo, Japan).

## Pfam domain search

Pfam domains were detected from amino acid sequences using Simple Modular Architecture Research Tool (SMART) (http://smart.embl-heidelberg.de/). Mouse FBXO24 amino acid sequence (CCDS51674.1) and human FBXO24 (CCDS5698.1) were obtained from the CCDS database (https://www.ncbi.nlm.nih.gov/CCDS/CcdsBrowse.cgi).

## Alignment of amino acid sequences

Amino acid sequences of mouse FBXO24 (CCDS51674.1) and human FBXO24 (CCDS5698.1) were aligned using 'Clustal Omega (https://www.ebi.ac.uk/Tools/msa/clustalo/)'. Aligned sequences were edited using Jalview version 2.11.2.0 (https://www.jalview.org/). Domains within the sequences were identified using 'SMART (http://smart.embl-heidelberg.de/)'.

## In silico expression data analysis

Single-cell transcriptome data in the mouse and human testis was obtained from previously published work (*Hermann et al., 2018*). *Fbxo24* expression in those cells was analyzed using the Loupe Cell Browser 3.3.1 (10X Genomics, Pleasanton, CA, USA).

## Plasmids

The open reading frames of *Fbxo24* and *Skp1* were cloned and amplified using cDNA obtained from mouse testis and inserted into the multiple cloning site of pCAG1.1 vector (Addgene; Plasmid #173685). To generate the expression vector coding *Fbxo24(ΔF-box)*, inverse PCR was performed with KOD -Plus- Mutagenesis Kit (Toyobo, Tokyo, Japan) according to the manufacturer's instructions. Primers used for inverse PCR are listed in *Supplementary file 1*. Human HA-Ubiquitin was obtained from the Addgene (plasmid # 18712; *Kamitani et al., 1997*).

## Observation of spermatozoa migration inside the female reproductive tract

The *Fbxo24* KO mouse line was crossed with B6D2 Tg mice carrying CAG/Su9-DsRed2, Acr3-EGFP (RBGS) (*Hasuwa et al., 2010*) to label the mitochondria with DsRed2. B6D2F1 females were super-ovulated by injecting pregnant mare serum gonadotropin (PMSG, ASKA Pharmaceutical, Tokyo, Japan) followed by human chorionic gonadotropin (hCG, ASKA Pharmaceutical) 48 hr apart. The superovulated female was mated with a WT male or *Fbxo24* KO male mice 12 hr after the hCG injection. Female mice were sacrificed 4 hr after confirming a vaginal plug and the female reproductive tracts were collected. Spermatozoa inside the oviducts were observed using a BZ-X710 microscope (Keyence Japan, Osaka, Japan).

## Protein extraction from testes, spermatozoa, and culture cells

HEK293T cells were lysed in 1% Triton X-100 lysis buffer (1% Triton X-100, 50 mM Tris-HCl pH 7.4, 150 mM NaCl, and 1% [vol/vol] protease inhibitor cocktail [Nacalai Tesque]) at 4°C with end-over-end rotation for 2 hr. Testes or spermatozoa were homogenized in 1% Triton X-100 lysis buffer at 4°C with end-over-end rotation for 2 hr or sample buffer containing 66 mM Tris-HCl pH 6.8, 2% SDS, 10% glycerol, and 0.005% Bromophenol Blue, and boiled for 5 min. Supernatants were obtained after centrifugation at 15,300×*g* for 15 min at 4°C.

## Immunoprecipitation

Immunoprecipitation was performed as described previously (*Kaneda et al., 2023*). Proteins from HEK293T cells were extracted using 1% Triton X-100 lysis buffer (1% Triton X-100, 50 mM Tris-HCl pH 7.4, 150 mM NaCl, and 1% [vol/vol] protease inhibitor cocktail [Nacalai Tesque]). The lysates were centrifuged at 15,300×*g* for 5 min at 4°C to collect supernatants. Protein lysates were mixed with 20 µl Protein G-conjugated magnetic beads (#10009D, Thermo Fisher Scientific) preincubated with 1.0 µg antibody. After incubation for 1 hr at 4°C, the beads were washed three times with wash buffer (40 mM Tris-HCl pH 7.5, 150 mM NaCl, 0.1% Triton X-100, and 10% glycerol). For immunoblotting, the immune complexes were eluted with 2×SDS sample buffer (132 mM Tris HCl pH 6.8, 4% SDS, 20% glycerol, and 0.01% Bromophenol Blue) for 10 min at 70°C. For MS, the immune complexes were eluted with 150 ng/µl 3×FLAG peptide (Sigma-Aldrich) in TBS solution for 30 min at 4°C. Antibodies used in this study are listed in *Supplementary file 2*.

## Immunoblotting

Proteins were separated with sodium dodecyl sulfate polyacrylamide gel electrophoresis (SDS-PAGE) under reducing conditions using 2-ME and transferred onto polyvinylidene difluoride membrane using the Trans Blot Turbo system (Bio-Rad, Hercules, CA, USA). After blocking with 10% skim milk (Becton Dickinson, Franklin Lakes, NJ, USA) in TBST, the blots were incubated with primary antibody overnight at 4°C, and then incubated with HRP-conjugated secondary antibodies (1:5000) for 2 hr at room temperature. Chemiluminescence was detected with Chemi-Lumi One Super (#02230, Nacalai Tesque) or Chemi-Lumi One Ultra (#11644, Nacalai Tesque).

## In vivo ubiquitination assay

HEK293T cells were transiently transfected with the plasmid DNA following the transfection protocol mentioned above. After 36 hr of transfection, HEK293T cells were treated with 10 µM MG132 for 9 hr. Immunoprecipitation and immunoblotting were performed following the abovementioned protocols.

## Assessment of sperm viability in vitro

Spermatozoa extracted from cauda epididymis were dispersed in TYH medium and incubated for 10 and 120 min at 37°C under 5% $CO_2$. PI was carefully added to the TYH drop to achieve a final concentration of 10 µg/ml and a small volume of the sperm suspension was placed on a glass slide. More than 200 spermatozoa were counted for each trial.

## Assessment of acrosome reaction in vitro

The *Fbxo24* KO mouse line was crossed with B6D2 Tg mice carrying CAG/Su9-DsRed2, Acr3-EGFP (RBGS) (*Hasuwa et al., 2010*) to label the acrosome with EGFP. Spermatozoa extracted from cauda

epididymis were dispersed in TYH medium for 10 and 120 min of incubation at 37°C under 5% $CO_2$. After 120 min of incubation, spermatozoa were incubated with $Ca^{2+}$ ionophore A23187 (Merck, Rahway, NJ, USA) at 20 μM to induce the acrosome reaction at 37°C under 5% $CO_2$. 10 mM stock solution of $Ca^{2+}$ ionophore A23187 in DMSO was diluted 10 times with TYH medium before carefully adding to the sperm suspension to reach the final concentration. PI (10 μg/ml at final concentration) was added to the TYH drop, and a small volume of the sperm suspension was placed on a glass slide. Acrosome-reacted spermatozoa were determined by observing EGFP signals while distinguishing viable spermatozoa with PI staining using a BX-53 microscope (Olympus). More than 200 spermatozoa were counted for each trial.

### Intracytoplasmic sperm injection
B6D2F1 females were superovulated by injecting PMSG (ASKA Pharmaceutical) followed by hCG (ASKA Pharmaceutical) 48 hr later. MII oocytes were collected 14 hr after injection of hCG. Cumulus cells were removed from the collected eggs after 10 min treatment with 330 μg/ml hyaluronidase (Sigma-Aldrich). Cumulus-free oocytes were placed in KSOM medium at 37°C under 5% $CO_2$ until just before performing ICSI. Each sperm head separated from the tail by applying a few piezo pulses was injected into a cumulus-free MII oocyte using a piezo manipulator (PrimeTech, Tokyo, Japan). Obtained two-cell embryos were transferred to pseudopregnant ICR females the next day.

### Immunofluorescence of testis cross sections
Testes were fixed in 4% paraformaldehyde in PBS for 3 hr at 4°C soon after dissection and infused with 15% and 30% sucrose in PBS at 4°C. The testes were embedded in OCT medium (Tissue-Tek, Miami, FL, USA) and frozen in liquid nitrogen. Testis blocks were sectioned at 10 μm thickness using a cryostat and the sections were dried on microscope slides at 42°C to make them adhere. The testis sections were then processed and imaged in the same manner as sperm immunostaining. After permeabilization with 0.1% Triton X-100 in PBS for 15 min, the sections were blocked with 3% bovine serum albumin (Sigma-Aldrich), 10% goat serum, and 0.1% Triton X-100 in PBS for 1 hr at room temperature. The sections were then incubated with primary antibody overnight at 4°C. After washing the sections with PBS for 10 min, the sections were secondarily blocked with 3% bovine serum albumin (Sigma-Aldrich), and 10% goat serum in PBS for 1 hr at room temperature. The sections were further incubated with fluorophore-conjugated secondary antibodies (1:2000) and Alexa Fluor 568 Dye PNA Lectin (1:2000) for 2 hr at room temperature. After a wash with PBS, the sections were incubated with Hoechst 33342 (Thermo Fisher Scientific) in PBS (2 μg/ml) for 5 min at room temperature. Once washed with PBS, the sections were mounted with Shandon Immu-Mount (Thermo Fisher Scientific) before imaging.

### Immunostaining of spermatozoa
Spermatozoa extracted from cauda epididymis were dispersed in PBS. A small aliquot of the sperm suspension was spotted onto glass slides and then air-dried at room temperature. Spermatozoa were then fixed with 4% paraformaldehyde in PBS for 10 min and washed three times with PBS for 5 min each. The slides were blocked with blocking solution (5% bovine serum albumin and 10% goat serum in PBS) for 1 hr at room temperature. The slides were incubated with primary antibody overnight at 4°C, washed with PBS three times for 5 min each, and incubated with Alexa Fluor-conjugated secondary antibody for 2 hr at room temperature. The slides were washed with PBS three times for 5 min each, incubated with Hoechst 33342 (2 μg/ml) (Thermo Fisher Scientific) for 5 min, and washed with PBS three times for 5 min each. The slides were mounted with Shandon Immu-Mount (Thermo Fisher Scientific) before imaging. Fluorescence images were captured with an Olympus BX-53 microscope (Olympus).

### Immunofluorescence of COS7 cells
COS7 cells were fixed with 4% paraformaldehyde in PBS for 15 min at room temperature. After permeabilization with 0.1% Triton X-100 in PBS for 15 min, cells were blocked with 1% bovine serum albumin (Sigma-Aldrich) in PBS for 1 hr at room temperature. The cells were then incubated with primary antibody overnight at 4°C. After washing three times with PBS, the cells were further incubated with fluorophore-conjugated secondary antibodies (1:1000) for 2 hr at room temperature. After

washing three times with PBS, the cells were incubated with Hoechst 33342 (Thermo Fisher Scientific) in PBS (2 μg/ml) for 5 min at room temperature. After washing three times with PBS, the cells were mounted with Shandon Immu-Mount (Thermo Fisher Scientific) before imaging.

## Extraction of RNA from epididymal spermatozoa

A small cut was made in the cauda epididymis with ophthalmic scissors and spermatozoa were squeezed out from the cut. Collected spermatozoa were dispersed in PBS and the number of spermatozoa was counted using a hemacytometer. Spermatozoa were centrifuged at 15,300×*g* for 2 min and the sperm pellet was lysed in TRIzol (Thermo Fisher Scientific). Chloroform was added to the lysate and incubated for 5 min at room temperature. The sample was centrifuged at 15,300×*g* for 15 min and the aqueous phase was transferred to a fresh centrifuge tube. To precipitate RNA, an equal volume of isopropanol was added to the aqueous phase and centrifuged at 15,300×*g* for 20 min at 4°C. The supernatant was removed and the pellet was washed with 75% ethanol. The washed pellet was dried and resuspended in nuclease-free water. The RNA solution was DNAse treated using deoxyribonuclease (RT Grade) (NIPPON GENE, Tokyo, Japan) to remove genomic DNA. Then the RNA solution was incubated at 55°C for 10 min. The sample was stored at –80°C. The concentration was measured using NanoDrop One (Thermo Fisher Scientific). The TAE/formamide method (*Masek et al., 2005*) was used to perform electrophoresis of RNA.

## Generation of *Fbxo24-FLAG* Tg mice

Two-PN embryos (B6D2 background) were obtained by IVF. The linearized DNA (*Fbxo24-FLAG* with *Prm1* promoter) was injected into one of the pronuclei. The embryos were cultivated and transferred into females as mentioned above. Genotyping was performed with primers (*Fbxo24* Tg Fw and *Fbxo24* Tg Rv) shown in *Supplementary file 1*.

## Statistical analysis

All statistical analyses were performed using the two-tailed Welch's t-test using Microsoft Office Excel 2016 (Microsoft Corporation, Redmond, WA, USA). Differences were considered significant at $p<0.05$ (*), $p<0.01$ (**), $p<0.001$ (***). Error bars are standard deviation.

## Acknowledgements

The authors would like to thank Dr. Julio M Castaneda and Ferheen Abbasi for the critical reading of the manuscript and Natsuki Furuta for technical assistance, and the members of both the Department of Experimental Genome Research and the nonprofit organization (NPO) for Biotechnology Research and Development for experimental assistance and discussion. We also thank Dr. Yoichi Miyamoto for kindly providing antibodies against importins (National Institutes of Biomedical Innovation, Health and Nutrition), Hiroko Omori for ultrastructural analysis, and Dr. Akinori Ninomiya and Dr. Fuminori Sugihara for MS analysis (Core Instrumentation Facility, Research Institute for Microbial Diseases, Osaka University).

## Additional information

### Funding

| Funder | Grant reference number | Author |
| --- | --- | --- |
| Japan Society for the Promotion of Science | JP23KJ1523 | Yuki Kaneda |
| Japan Society for the Promotion of Science | JP22H03214 | Haruhiko Miyata |
| Japan Society for the Promotion of Science | JP23K18328 | Haruhiko Miyata |
| Japan Society for the Promotion of Science | JP23K05831 | Keisuke Shimada |

| Funder | Grant reference number | Author |
|---|---|---|
| Japan Society for the Promotion of Science | JP22K15030 | Maki Kamoshita |
| Japan Society for the Promotion of Science | JP19H05750 | Masahito Ikawa |
| Japan Society for the Promotion of Science | JP21H04753 | Masahito Ikawa |
| Japan Society for the Promotion of Science | JP21H05033 | Masahito Ikawa |
| Takeda Science Foundation | | Haruhiko Miyata Keisuke Shimada Masahito Ikawa |
| Fusion Oriented REsearch for disruptive Science and Technology | JPMJFR211F | Haruhiko Miyata |
| Eunice Kennedy Shriver National Institute of Child Health and Human Development | P01HD087157 | Masahito Ikawa |
| Eunice Kennedy Shriver National Institute of Child Health and Human Development | R01HD088412 | Masahito Ikawa |
| Bill and Melinda Gates Foundation | INV-001902 | Masahito Ikawa |

The funders had no role in study design, data collection and interpretation, or the decision to submit the work for publication.

## Author contributions

Yuki Kaneda, Conceptualization, Resources, Formal analysis, Funding acquisition, Validation, Investigation, Writing - original draft, Writing - review and editing; Haruhiko Miyata, Masahito Ikawa, Conceptualization, Resources, Formal analysis, Supervision, Funding acquisition, Validation, Investigation, Writing - original draft, Project administration, Writing - review and editing; Zoulan Xu, Formal analysis, Investigation, Writing - review and editing; Keisuke Shimada, Formal analysis, Funding acquisition, Investigation, Writing - review and editing; Maki Kamoshita, Resources, Formal analysis, Funding acquisition, Investigation, Writing - review and editing; Tatsuya Nakagawa, Chihiro Emori, Resources, Formal analysis, Investigation, Writing - review and editing

## Author ORCIDs

Yuki Kaneda ⓘ https://orcid.org/0000-0001-6300-4234
Haruhiko Miyata ⓘ https://orcid.org/0000-0003-4758-5803
Keisuke Shimada ⓘ https://orcid.org/0000-0003-3739-7163
Chihiro Emori ⓘ https://orcid.org/0000-0001-8040-2517
Masahito Ikawa ⓘ https://orcid.org/0000-0001-9859-6217

## Ethics

All animal experiments were approved by the Animal Care and Use Committee of the Research Institute for Microbial Diseases, Osaka University (#Biken-AP-H30-01 and #Biken-AP-R03-01).

Reviewer #1 (Public review): https://doi.org/10.7554/eLife.92794.3.sa1
Author response https://doi.org/10.7554/eLife.92794.3.sa2

# Additional files

## Supplementary files

• Supplementary file 1. Primers and gRNAs used in this study.

- Supplementary file 2. Antibodies used in this study.
- MDAR checklist

## Data availability

All data generated or analyzed during this study are included in the manuscript and supporting files.

The following previously published dataset was used:

| Author(s) | Year | Dataset title | Dataset URL | Database and Identifier |
|---|---|---|---|---|
| Hermann BP, Cheng K, Singh A, Roa-De La Cruz L, Mutoji KN, Chen IC, Gildersleeve H, Lehle JD, Mayo M, Westernströer B, Law NC, Oatley MJ, Velte EK, Niedenberger BA, Fritze D, Silber S, Geyer CB, Oatley JM, McCarrey JR | 2018 | Queryable single-cell RNA-seq (Drop-Seq) datasets of Human and Mouse spermatogenic cells | https://doi.org/10.17632/kxd5f8vpt4.1 | Mendeley Data, 10.17632/kxd5f8vpt4.1 |

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
