## [Editor Report · eLife assessment]

This **important** study reports that FBXO24 is essential for the normal formation and function of the sperm flagellum, motility, and male fertility in mice. The evidence supporting the direct role of this protein in preventing RNP granule formation in the sperm flagellum is **compelling**. This work will be of interest to biomedical researchers who work on testicular biology and male fertility.

---

## [Referee Report · Reviewer #1 (Public review)]

Summary:

The main goal of the authors was to study the testis-specific role of the protein FBXO24 in the formation and function of the ribonucleoprotein granules (membrane-less electron-dense structures rich in RNAs and proteins).

Strengths:

The wide variety of methods used to support their conclusions (including transgenic models)

Weaknesses:

The complex phenotype observed, in some situations, cannot be fully explained by the experiments presented by the authors (i.e., AR or the tail structure).

---

## [Author Response]

The following is the authors’ response to the original reviews.

**Public Reviews:**

**Reviewer #1 (Public Review):**
Summary:The main goal of the authors was to study the testis-specific role of the protein FBXO24 in the formation and function of the ribonucleoprotein granules (membraneless electron-dense structures rich in RNAs and proteins).

We appreciate the summary comment of reviewer #1.

Strengths:The wide variety of methods used to support their conclusions (including transgenic models)

We appreciate the positive comment of reviewer #1.

Weaknesses:The lack of specific antibodies against FBXO24. Some of the experiments showing a specific phenotype are descriptive and lack of logical explanation about the possible mechanism (i.e. AR or the tail structure).

Because we could not obtain specific antibodies against FBXO24, we generated Fbxo24-FLAG transgenic mice, which can be used to show the interaction between FBXO24 and IPO5. For the mechanism of impaired acrosome reaction, we added some results and discussion as written in the response to the question (1) of reviewer #1 (public review). For the mechanism of abnormal flagellar structure, we added new results and fixed the manuscript as written in the response to the major comments of reviewer #3 (recommendations for the authors).

Questions:The paper is excellent and employs a wide variety of methods to substantiate the conclusions. I have very few questions to ask:(1) KO mice cannot undergo acrosome reaction (AR) even spontaneously. How do you account for this, given that no visible defects were observed in the acrosome?

One possibility is that Fbxo24 KO spermatozoa cannot undergo capacitation; however, it is difficult to analyze the capacitation status such as tyrosine phosphorylation because most Fbxo24 KO spermatozoa are not alive (Figure S3A). Other possibility is that AR-related proteins are affected in Fbxo24 KO spermatozoa. Therefore, we analyzed the amounts of AR-related proteins with mass spectrometry (Figure S3C). Although previous studies indicate that the assembly of the SNARE complex is a key event prior to AR [Hutt et al., 2005 (PMID: 15774481); Katafuchi et al., 2000 (PMID: 11066067); Schulz et al., 1997 (PMID: 9356173); Tomes et al., 2002 (PMID: 11884041)], no clear differences were detected for SNARE proteins (Figure S3C and D). PLCD4 that is important for AR Fukami et al., 2001 (PMID: 11340203) was also detected in Fbxo24 KO spermatozoa (Figure S3C). Although we could not find differences in the amounts of AR-related proteins, it is still possible that FER1L5, another AR-related protein [Morohoshi et al., 2023 (PMID: 36696506)] not detected in the mass spectrometry analyses, or AR-related proteins not yet identified are affected in Fbxo24 KO spermatozoa. We added these results and discussion (line 160-166 and 305-312).

(2) KO sperm are unable to migrate in the female tract, and, more intriguingly, they do not pass through the utero-tubal junction (UTJ). The levels of ADAM3 are normal, suggesting that the phenotype is influenced by other factors. The authors should investigate the levels of Ly6K since mice also exhibit the same phenotype but with normal levels of ADAM3.

We detected LY6K in Fbxo24 KO spermatozoa with immunoblotting, but no difference was found.

We added the results (Figure S3E and line 172–175).

(3) In Figure 4A, the authors assert that "RBGS Tg mice revealed that mitochondria were abnormally segmented in Fbxo24 KO spermatozoa." I am unable to discern this from the picture shown in that panel. Could you please provide a more detailed explanation or display the information more explicitly?

We are sorry for the ambiguous explanation on the morphology of sperm mitochondria sheath. Fbxo24 KO cauda epidydimal spermatozoa shows disorganized mitochondria sheath rather than “segmented”. We fixed the sentence (line 190-192) and added white arrowheads that indicate the disorganized regions (Figure 4A).

**Reviewer #2 (Public Review):**
Summary:The manuscript by Kaneda et al "FBXO24 ensures male fertility by preventing abnormal accumulation of membraneless granules in sperm flagella" is a significant paper on the role of FBXO24 in murine male germ cell development and sperm ultrastructure and function. The body of experimental evidence that the authors present is extraordinarily strong in both breadth and depth. The authors investigate the protein's functions in male germ cells and sperm using a wide variety of approaches but focusing predominantly on their novel mouse model featuring deletion of the Fbxo24 gene and its product. Using this mouse, and a cross of it with another model that expresses reporters in the head and midpiece, they logically build from one experiment to the next. Together, their data show that this protein is involved in the regulation of membraneless electron-dense structures; loss of FBXO24 led to an accumulation of these materials and defects in the sperm flagellum and fertilizing ability. Interestingly, the authors found that several of the best-known components of electron-dense ribonucleoprotein granules that are found in the intermitochondrial cement and chromatoid body were not disrupted in the Fbxo24 knockout, suggesting that the electron-dense material and these structures are not all the same, and the biology is more complicated than some might have thought. They found evidence for the most changes in IPO5 and KPNB1, and biochemical evidence that FBXO24 and IPO5 could interact.

We appreciate the summary comment of reviewer #2.

Strengths:The authors are to be commended for the thoroughness of their experimental approaches and the extent to which they investigated impacts on sperm function and potential biochemical mechanisms. Very briefly, they start by showing that the Fbxo24 message is present in spermatids and that the protein can interact with SKP1, in a way that is dependent on its F-box domain. This points toward a potential function in protein degradation. To test this, they next made the knockout mouse, validated it, and found the males to be sterile, although capable of plugging a female. Looking at the sperm, they identified a number of ultrastructural and morphological abnormalities, which they looked at in high resolution using TEM. They also cross their model with RBGS mice so that they have reporters in both the acrosome and mitochondria. The authors test a variety of sperm functions, including motility parameters, ability to fertilize by IVF, cumulus-free IVF, zona-free-IVF, and ICSI. They found that ICSI could rescue the knockout but not other assisted reproductive technologies. Defects in male fertility likely resulted from motility disruption and failure to get through the utero-tubal junction but defects in acrosome exocytosis also were noted. The authors performed thorough investigations including both targeted and unbiased approaches such as mass spectrometry. These enabled them to show that although the loss of the FBXO24 protein led to more RNA and elevated levels of some proteins, it did not change others that were previously identified in the electron-dense RNP material.The manuscript will be highly significant in the field because the exact functions of the electron-dense RNP materials have remained somewhat elusive for decades. Much progress has been made in the past 15 years but this work shows that the situation is more complex than previously recognized. The results show critical impacts of protein degradation in the differentiation process that enables sperm to change from non-descript round cells into highly polarized and compartmentalized mature sperm, with an equally highly compartmentalized flagellum. This manuscript also sets a high bar for the field in terms of how thorough it is, which reveals wide-ranging impacts on processes such as mitochondrial compaction and arrangement in the midpiece, the correct building of the major cytoskeletal elements in the flagellum, etc.

We appreciate the positive comment of reviewer #2.

Weaknesses:There are no real weaknesses in the manuscript that result from anything in the control of the authors. They attempted to rescue the knockout by expressing a FLAG-tagged Fbxo24 transgene, but that did not rescue the phenotype, either because of inappropriate levels/timing/location of expression, or because of interference by the tag. They also could not make anti-FBXO24 that worked for coimmunoprecipitation experiments, so relied on the FLAG epitope, an approach that successfully showed co-IP with IPO5 and SKP1.

We could not rescue the phenotype with Fbxo24-FLAG transgene, but different Fbxo24 mutant mice show the same phenotypes (Figure S6G). Further, another group showed that Fbxo24 KO mice exhibited abnormal mitochondrial coiling [Li et al., 2024 (PMID: 38470475)], confirming that

FBXO24 is involved in the mitochondrial sheath formation.

**Reviewer #3 (Public Review):**
Summary:In this manuscript, the authors found that FBXO24, a testis-enriched F-box protein, is indispensable for male fertility. Fbxo24 KO mice exhibited malformed sperm flagellar and compromised sperm motility.

We appreciate the summary comment of reviewer #3.

Strengths:The phenotype of Fbxo24 KO spermatozoa was well analyzed.

We appreciate the positive comment of reviewer #3.

Weaknesses:The authors observed numerous membraneless electron-dense granules in the Fbxo24 KO spermatozoa. They also showed abnormal accumulation of two importins, IPO5 and KPNB1, in the Fbxo24 KO spermatozoa. However, the data presented in the manuscript do not support the conclusion that FBXO24 ensures male fertility by preventing the abnormal accumulation of membraneless granules in sperm flagella, as indicated in the manuscript title.

Fbxo24 KO mice showed abnormal accumulation of membraneless granules in sperm flagella and male infertility, suggesting that FBXO24 is involved in these processes, but there are no results that show the direct relationship as reviewer #3 mentioned. Therefore, we fixed the title.

**Recommendations For The Authors:**

**Reviewer #2 (Recommendations For The Authors):**
On page 4, lines 152-154, the authors introduce the RBGS mouse model and use it in their experiments.However, they left out an obvious but helpful sentence that tells the reader that they crossed the Fbxo24-null mouse with the RBGS. As one continues reading it is clear, but best to avoid even slight confusion.

We revised the explanation in the result section (line 150-153).

**Reviewer #3 (Recommendations For The Authors):**
In this manuscript, the authors found that FBXO24, a testis-enriched F-box protein, is indispensable for male fertility. Fbxo24 KO mice exhibited malformed sperm flagellar and compromised sperm motility. The phenotype of Fbxo24 KO spermatozoa was well analyzed.The authors observed numerous membraneless electron-dense granules in the Fbxo24 KO spermatozoa. They also showed abnormal accumulation of two importins, IPO5 and KPNB1, in the Fbxo24 KO spermatozoa. However, the data presented in the manuscript do not support the conclusion that FBXO24 ensures male fertility by preventing the abnormal accumulation of membraneless granules in sperm flagella, as indicated in the manuscript title.

Fbxo24 KO mice showed abnormal accumulation of membraneless granules in sperm flagella and male infertility, suggesting that FBXO24 is involved in these processes, but there are no results that show the direct relationship as reviewer #3 mentioned. Therefore, we fixed the title.

Major comments:In the title, abstract, introduction, and some sections such as lines 275-276, the authors conclude that FBXO24 prevents the accumulation of importins and RNP granules during spermiogenesis. However, the provided data do not substantiate this claim. To provide conclusive evidence to support the current title, the authors need to present evidence supporting: (1) direct degradation of IPO5 and KPNB1 by FBXO24; (2) the direct requirement of IPO5 for the formation of the membraneless granules, and (3) infertility resulting from the presence of membraneless granules, rather than other issues such as abnormal ODF and AX.

(1) direct degradation of IPO5 and KPNB1 by FBXO24.

To examine if IPO5 can be degraded by FBXO24, we performed a ubiquitination assay using HEK293T cells. Ubiquitination of IPO5 was upregulated in the presence of WT FBXO24 but not with the mutant ΔF-box FBXO24, suggesting that IPO5 can be ubiquitinated by FBXO24. We did not examine the ubiquitination of KPNB1 because we failed to construct a plasmid vector expressing mouse KPNB1. We think that KPNB1 is not the substrate because we did not detect the interaction between FBXO24 and KPNB1 (Figure 5E). We added the results of the ubiquitination assay (Figure 5F and line 261-265) and mentioned it in the abstract (line 35).

(2) the direct requirement of IPO5 for the formation of the membraneless granules.

(3) infertility resulting from the presence of membraneless granules, rather than other issues such as abnormal ODF and AX.

We revealed that IPO5 aggregate under stress condition in COS7 cells (Figure 6C and D); however, we did not examine whether IPO5 is required for the formation of the membraneless granules. We consider that protein degradation systems such as PROTAC or Trim-Away to knockdown IPO5 at the protein level in Fbxo24 KO mice could be a good way to see if the membraneless granules are diminished and male fertility is rescued. However, it takes time to apply the degradation systems in vivo. Therefore, we would like to leave this rescue experiment for future studies. We fixed the title and abstract (line 37-38), and removed the last sentence of the introduction.

Also, the other group reported the analyses of Fbxo24 KO mice [Li et al., 2024 (PMID: 38470475)] right after we submitted our manuscript to the eLife. They reported not only disorganized flagellar structures but also abnormal head morphology, which may lead to male infertility. The differences from our study may be due to different mouse genetic backgrounds. We mentioned it in the discussion section (line 348-353).

Minor comments:(1) The authors claimed a significant increase in the total amount of RNAs in Fbxo24 KO spermatozoa (lines 259-261), suggesting that the ...contain RNAs. More direct evidence supporting this claim should be provided.

We show that the amounts of IPO5 and KBNB1 increased in Fbxo24 KO spermatozoa (Figure 5A and B), both of which could be incorporated into RNP granules in COS7 cells (Figure 6C and D), supporting the idea that membraneless electron-dense structures may be RNP granules. However, because we did not show direct evidence that electron-dense structures contain RNAs, we removed the sentences (line 259-261 of the 1st submission manuscript).

(2) The author should provide an explanation for the absence of a FLAG band in the input Tg in Figure 5D and the larger size of the IPO5 band in the FLAG-IP group compared to the input. Similar observations are also noted in Figure 5E.

The FLAG band is weak because the protein amount is low. When we increase the contrast, we can see the FLAG band. We added an image with high contrast (Figure 5D). Sometimes, proteins run differently with SDS-PAGE after immunoprecipitation, likely due to varying protein composition in the sample. We explained it in the figure legend (line 868-869).

(3) In Line 526, clarify the procedure for sperm purification, and determine the potential for contamination from somatic cells.

We did not perform sperm purification, but when we observed spermatozoa obtained from cauda epididymis, we rarely observed either somatic cells or immature spermatogenic cells. We added pictures in Figure S7. Further, we added detailed explanation about how to collect spermatozoa from the epididymis (line 549-550).

(4) Define the Y-axis in Figure 2E, F, and G.

We have revised the figures.